

# Ice-core data used for the construction of the Greenland Ice-Core Chronology 2005 and 2021 (GICC05 and GICC21)

Sune Olander Rasmussen[1], Dorthe Dahl-Jensen[1,2], Hubertus Fischer[3], Katrin Fuhrer[3,4], Steffen Bo Hansen[1], Margareta Hansson[5], Christine S. Hvidberg[1], Ulf Jonsell[6], Sepp Kipfstuhl[7], Urs Ruth[7,8], Jakob Schwander[3], Marie-Louise Siggaard-Andersen[9], Giulia Sinnl[1], Jørgen Peder Steffensen[1], Anders M. Svensson[1], and Bo M. Vinther[1]

[1]Centre for Ice and Climate, Section for the Physics of Ice, Climate, and Earth, Niels Bohr Institute, University of Copenhagen, Copenhagen, DK-2200, Denmark
[2]Centre for Earth Observation Science, University of Manitoba, Winnipeg, MB R3T 2N2, Canada
[3]Climate and Environmental Physics, Physics Institute, and Oeschger Center for Climate Change Research, University of Bern, Bern, Switzerland
[4]TOFWERK AG, CH-3645 Thun, Switzerland
[5]Department of Physical Geography, University of Stockholm, Stockholm, Sweden
[6]Swedish Research Council, Stockholm, 112 21, Sweden
[7]Alfred Wegener Institute Helmholtz Center for Polar and Marine Science, Bremerhaven, Germany
[8]Robert Bosch GmbH, Corporate Research, Stuttgart, 70049, Germany
[9]Section for Geogenetics, Globe Institute, University of Copenhagen, Copenhagen, Denmark

*Correspondence to*: Sune Olander Rasmussen (sune.rasmussen@nbi.ku.dk)

**Abstract.** We here describe, document, and make available a wide range of data sets used for annual layer identification in ice cores from DYE-3, GRIP, NGRIP, NEEM, and EGRIP. The data stem from detailed measurements performed both on the main deep cores and shallow cores over more than forty years using many different setups developed by research groups in several countries, and comprise both discrete measurements from cut ice samples and continuous-flow analysis data.

The data series were used for the construction of the Greenland Ice-Core Chronology 2005 (GICC05) and/or the revised GICC21. Now that the underlying data are made available, we also release the individual annual layer positions of the GICC05 time scale which are based on these data sets.

We hope that the release of the data sets will stimulate further studies of the past climate taking advantage of these highly resolved data series covering a large part of the interior of the Greenland ice sheet.

## 1 Introduction

The full potential of palaeoclimatic data relies on a reliable time scale, i.e., a depth−age relation, and identification and counting of annual-layers is the most accurate way to obtain a time scale if high-resolution measurements of parameters showing annual variability are available. In Greenland, the interglacial surface accumulation rate is ~0.1-0.5 m of ice equivalent per year in



interior areas where the deep ice cores are drilled, thereby allowing annual layers to be identified in the Holocene period and into the last glacial. Not all parameters have been measured along all the entire ice cores, and the resolving power of measurements depends both on the measurement resolution and the annual-layer thickness, which vary due to past climate changes as well as layer thinning and generally decrease with depth, so annual-layer counting is often only possible within a certain interval for each combination of parameter and ice core. A particular challenge is the so-called brittle zone, where the pressure release and temperature rise experienced by the cores after drilling cause internal cracking of the ice, making uncontaminated continuous measurements of the ice difficult. The brittle zone is found in Greenland at depths around 800-1400 meters (sometimes setting in earlier), where the air bubbles are compressed under pressure and gradually transformed into clathrate hydrates (Kipfstuhl et al., 2001).

The Greenland Ice-Core Chronology (GICC) is an attempt to derive a consistent, common time scale for the Greenland ice cores by combining data from multiple cores, using for each time period all available annually resolved data and then applying the time scale to the other cores by means of matching patterns in volcanic and other non-climatic events. In this way, data from all the ice cores can be interpreted together on a common time scale (i.e., with very small relative dating uncertainty), greatly reducing the risk of artificial offsets due to misinterpretation of individual records. The first sections of GICC were published in 2006 and cover the time interval from present day back to 14.8 ka b2k (thousand years before 2000 CE) (Vinther et al., 2006; Rasmussen et al., 2006), and the dating was continued in the glacial back to 42 ka b2k (Andersen et al., 2006) and onwards to 60 ka b2k (Svensson et al., 2008), at which point the layers had thinned too much to continue with continuous annual layer counting. The time scale, named GICC05 because the first manuscripts were submitted in 2005, was therefore extended with a flow-model based time scale to cover the remaining glacial period (Wolff et al., 2010). Since then, data from the newer Greenland ice cores NEEM and EGRIP have appeared, and comparisons to other time scales have shown that GICC05 was not as accurate as initially assumed (Sigl et al., 2013), and in 2021, the revision of GICC05 was started by Sinnl et al. (2022), producing the revised GICC21 time scale covering the most recent 3.8 ka using data from many parallel ice cores. Some of the data sets used for GICC05 and GICC21 are publicly available, but far from all. With the advent of the revised GICC21 time scale, some of the data sets have been used again, and others will be used as the revision proceeds, and this calls for all data files being made publicly available. Also, the actual time scale was only released in 20-yr resolution mainly because the Holocene isotope data series and the NGRIP chemistry data sets, on which most of the annual layer identification was based, were not publicly available at that time. These data sets are now being made available here or in the recent paper by Erhardt et al. (2022), and it thus seems timely also to make the fully resolved annual layer positions available.

The data files span several decades of work, come from a range of analysis methods, and are related to different ice cores. Below, we first introduce the drill locations and then go through the different types of measurements.



## 2 Ice-coring locations


Data from the DYE-3, GRIP, NGRIP, NEEM, and EGRIP cores are presented here (see Fig. 1 for locations). In some cases, there are several cores from each drill site.

### 2.1 The DYE-3 cores

The DYE-3 deep ice core was drilled to bedrock near the US radar station from which the core takes its name in 1979–1981

during the American–Danish–Swiss Greenland Ice Sheet Program, also known as GISP (Dansgaard et al., 1982). The position of camp is often given as 65°N, 44°W, but notes from the time of drilling provide the more precise position 65.18°N, 43.83°W with an altitude of 2480 m above sea level. At the time of drilling, the mean annual surface temperature was determined to –19°C, while bore-hole thermometry produces a modern temperature of –20°C (Dahl-Jensen et al., 1998). The ice core retrieved by 1981 was 2035 m long, and at the end of the season, the drill was stuck. The drill was recovered in 1982 with the last 2

meters of core. Almost 90% of the ice originates from the Holocene, and the brittle zone included the section between 800 and 1400 m depth. Present-day accumulation is on the order of half a meter, but variable due to upstream surface undulations (Vinther et al., 2006). Despite the relatively thin glacial section, the analysis of oxygen isotopes resolved and confirmed the same repeated abrupt glacial climatic changes previously found in the Camp Century ice core drilled 1400 km away, which we today call the Dansgaard−Oeschger Events.

The location of the DYE-3 drill site was determined by the location of the radar station and is not the ideal place to drill an ice core, as flow of ice from upstream areas over a mountainous bedrock and 41.5 km from the ice divide complicates the interpretation. As part of the GISP activities, shallow cores as well as velocity and altitude measurements were made every 2 km along a line upstream from the DYE-3 station (the B line), and along two parallel lines offset by 2 km to the NNW (A line) and SSE (C line) in order to better understand the flow of ice leading to the DYE-3 site (see Whillans et al. (1984), for details)

and to correct for anomalous thinning of the annual layers due to the upstream bedrock (Reeh et al., 1985). The shallow core DYE-3 4B was drilled to 174 m depth 8 km upstream from DYE-3, and shallow core DYE-3 18C was drilled to 110 m depth 36 km upstream along the C line at the position 65.03°N 44.39°W.

### 2.2 The GRIP core

The core from GRIP (Greenland Ice-Core Project) was drilled during the 1989−1992 field seasons near the highest point of

the Greenland ice sheet, Summit (72.57°N 37.62°W, 3232 m above sea level). The core length is 3028.8 m, and the present-day accumulation is 0.23 m ice/yr (Johnsen et al., 1992; Dansgaard et al., 1993; Dahl-Jensen et al., 1993) and the surface temperature was measured to -32°C (Hvidberg et al., 1997). The core is drilled close to the present-day ice divide, and the ice divide has a low enough slope to allow us to ignore upstream corrections, although we acknowledge that the ice-flow configuration could have been different in the past. Indeed, changing ice-flow configurations during the last glacial cycle are

thought to have altered the bottom 200 m of the GRIP core by folding, identified by Grootes et al. (1993) as inconsistencies



with the neighbour GISP2 ice core drilled 28 km away. In the top 101.3 m, measurements were performed on a shallow core (the S3 shallow core) drilled close to the main hole, but the S3 data are fully integrated in main core records, so in practice, the GRIP record is treated like coming from one core.

## 2.3 The NGRIP cores

Drilling of the NGRIP ice core (Dahl-Jensen et al., 2002) was successfully completed in 2003 when liquid water was found at the bedrock at a depth of 3085 m at 75.10°N, 42.32°W. The drill site had an elevation of 2,917 m and a mean temperature of -31.5°C during the years where the camp was operational. The melting at the base limits the age of the ice in the core to approx. 123 ka b2k (North Greenland Ice Core Project Members, 2004) and probably for the same reason, folding at the bottom has not been observed. A 45-m long replicate core from the deepest section was drilled in 2004 and goes 6 m deeper, but data from

this core are not presented here. The present-day accumulation rate is 0.19 m of ice/yr and the bottom melting results in a flow pattern different from that of GRIP with slow flow of approximately 1 m/yr along the ice ridge and less thinning of the bottom layers due to the high basal melt rate of several mm/yr (Buchardt and Dahl-Jensen, 2007). Thus, the annual layers are more than 5 mm thick over the entire length of the NGRIP core, making it ideal for annual-layer identification.

The NGRIP record comes from a combination of two ice cores: the drill got stuck in 1997 and a new core had to be drilled.

The two cores are referred to as NGRIP1 and NGRIP2, respectively, and measurements have been performed on the NGRIP1 core down to a depth of 1372 m, while measurements on the NGRIP2 core start at a depth of 1346 m (corresponding to approximately 9.5 ka b2k) with an overlap (below the brittle zone) to ensure correct alignment of the two records. The mean offset of similar features seen in both the NGRIP1 and NGRIP2 cores is 0.43 m, with the same feature appearing at greater depths in the NGRIP1 core than in the NGRIP2 core (Hvidberg et al., 2002), which is opposite to what would be expected

given that the NGRIP2 cores was started several years later. The offset was found to be mainly caused by accumulated uncertainties in the logging of the NGRIP1 core across the brittle section.

## 2.4 The NEEM cores

A 2,540-m-long ice core was drilled during 2008–2012 through the ice at the NEEM site, Greenland (77.45°N, 51.06°W, surface elevation 2,450 m, mean annual temperature –29°C, annual accumulation rate 22 cm of ice equivalent). See NEEM

Community Members (2013) for more details of the drilling and NEEM ice core and Rasmussen et al. (2013) for a description of the NEEM dating efforts. A 400 m long shallow core named NEEM-2011-S1 was drilled about 100 m away from the NEEM main core drill site (Sigl et al., 2013).

## 2.5 The EGRIP core

The EGRIP, or EastGRIP, ice core is at the time of writing (2022) still being drilled in NNE Greenland. At the start of the

drilling operation in 2016, the drilling site was located at 75.6°N, 36.0°, and moves annually more than 50 m towards the NE together with the Northeast Greenland Ice Stream. The present-day temperature is –30°C and the average annual accumulation

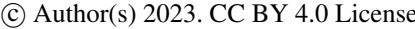

rate is equivalent to 0.11 m of ice, determined as the average over the period 1607–2011 from a firn core close to the main EGRIP drilling site (Vallelonga et al., 2014) and confirmed later by annual layer counts on the main core. The upper part of the EGRIP core was first dated by transferring GICC05 to the core by matching of mainly volcanic markers (Mojtabavi et al., 2020; Gerber et al., 2021). Later, the EastGRIP records were included in the GICC revision, leading to a revised time scale for the past 3.8 ka (Sinnl et al., 2022).

## 3 Analysis methods

We here provide information on the measurement methods used for the presented ice-core records. The information given is correct to the best of our knowledge, but considering that some of the data sets were produced decades ago using equipment that no longer exists by people who are no longer with us, we sometimes have to resort to estimated uncertainties. Uncertainties on individual methods will be discussed below, but we here discuss uncertainties in depth assignment which are particularly important when comparing data obtained on their own, different and independent depth scales. When an ice core has been drilled, it is logged, a process which involves establishing the master depth scale for the core (Hvidberg et al., 2002). Cores are drilled in segments of up to 4 meters, and in the vast majority of cases, the drilled cores match perfectly with the next core, enabling a highly precise depth assignment across core breaks (estimated uncertainty of 1 mm or less). Where drilling problems or core breaks have damaged the core, a larger uncertainty is introduced. This uncertainty will typically be in the order of a few millimetres in case of irregular breaks, but in a few cases where a piece of ice core has been lost in the drilling process or during extraction from the drill, the uncertainty can be larger. Outside the brittle zone, this is very rare. As this uncertainty applies to the master depth assignment, all measurements performed on the core will be affected in the same way, and for this reason, this is rarely a critical issue when interpreting data from the cores. However, it does introduce localized but large uncertainty in the derived annual layer thicknesses and could cause problems if the data are compared to data obtained from radar measurements (which rarely have sufficient resolution for this to be a significant problem, though), or data obtained directly in/from the bore-hole. To summarize, the master depth assignment is accurate and any uncertainties in the master depth assignment will apply to all measurements in the same way, not influencing the relative depth precision between different records from the same core. After logging, the core is split into sections for further analysis. From this point onward, the depth uncertainties will be different between different measurement systems or sampling methods, e.g.

- Some data series are measured directly on the core (e.g. DEP, ECM, visual stratigraphy). The instruments measure the location of the instrument/sensor along the core, which is often accurate (mm-scale uncertainties or less for DEP and visual stratigraphy), while the ECM setup has higher uncertainty because it is necessary to have some flexibility in how to move the electrodes across the surface, and due to nonlinearity of the potentiometer measuring the along-core position. From parallel measurements on the same core, we estimate the depth assignment uncertainty to be 0.5 cm at the ends and up to 2 cm at the middle of the core sections when making measurements across breaks where smooth operation of the electrodes is difficult.



- Some measurements are made on discrete samples cut from the core (the water stable isotopes reported here and some impurity data, e.g. the ion chromatography data presented below). Each data point will represent an average across a depth interval (often 5 cm minus the ~2 mm width of the saw cut), but the depth assignment (relative to the master depth scale) is very accurate, estimated to 1-2 mm, and given by the width of the sawblade and pencil markings.

- More recently, continuous measurements on melted samples have become common (see section 3.5). While the depth assignment of the end of each core section is accurate, assigning depths in the middle of a section relies on the assumed or measured melt speed. This introduces a possible depth uncertainty from the true (master) depth assignment which is likely similar for all measured species, but will produce an artificial offset compared to e.g. discretely measurements.

These fundamentally different uncertainty contributions will both rely on equipment-specific issues and operator care and experience. Our observations from multi-parameter data sets containing peaks that would be expected to align in ECM, DEP and impurity data show that relative, and probably artificial, offsets 2-3 cm are not uncommon, although data sets most often are aligned within ~1 cm. When interpreting data from parallel records from the same core, the observer should be aware of the limitations imposed by these uncertainties (which are on the same scale as the typical annual layer thickness, or smaller). Analysis of sub-annual leads and lags is thus not generally recommend unless special care is taken to improve the relative alignment of the records.

## 3.1 Electrical Conductivity Measurements (ECM)

ECM data (reflecting solid-state DC conductivity) were obtained with the technique described by Hammer (1980) and modern versions hereof. It was soon clear that the main value of ECM data was as a tool for aligning records using patterns of volcanic peaks, where absolute calibration plays a lesser role, and therefore not much work has gone into maintaining absolute calibration measurements for the more recent projects. During the DYE-3 drilling project, the method used a direct voltage of 1250 V, and the method was calibrated using pH measurements on discrete samples (Hammer, 1980). In later projects, the voltage was changed to 2000 V to enhance the signal, but not calibrated accurately at these new sites. The data are given as [H$^+$] versus depth (measured from the undisturbed surface at the year where drilling started), but as described, the absolute calibration must be considered tentative, also because the effect of core temperature has not been accounted for systematically. Furthermore, the measurement depends on the density of the ice core due to the experimental technique, so the calibration in the firn is not the same as in the ice.

Data from the DYE-3 cores (main and shallow cores) were originally recorded by an analogue plotter on paper in high resolution and later digitized by hand in 1 cm resolution by laboratory assistant Anita Boas. The initial paper plots were of high resolution, and we find it very likely that the depth uncertainty contribution arising from the digitization is negligible compared to the depth uncertainty related to the original depth assignement which – as decribed above – can be up to a few centimeters in the middle of each measured section. The concentration uncertainty contribution from the digitization is certainly also negligible compared to the uncertainty arising from the tentative calibration, and irrelevant when the ECM data



are used for matching of patterns of volcanic signals. Regarding calibration, see Hammer (1980), Hammer et al. (1985) and Neftel et al. (1985).

Data from GRIP were recorded in parallel on paper in high resolution and digitally in 1 cm resolution. The calibration applied
is $[H^+] = 0.045\ I^{1.73}$ µequiv./kg (where I is the current in µA), but it must be considered tentative (see Clausen et al. (1995)). ECM data from NGRIP1, NGRIP2, NEEM and EGRIP are publicly available. See section 4 for data sources.

### 3.2 Dielectrical Properties (DEP)

Unlinke ECM, Di-electric Profiling (DEP) is an AC (alternating current) method. It was developed by Moore and Paren (1987) and Moore et al. (1989) as a technique to determine the dielectric properties of snow and ice and the total ionic concentration
in ice cores at a time when direct measurements of ionic concentrations by Ion Chromatography (IC, section 3.4) and Continuous Flow Analysis (CFA, section 3.5) were not yet routinely applied. As a non-destructive method – the complete core is lying between curved electrodes forming a capacitor – DEP is usually the first measurement in an ice-core processing line. Classically, DEP measures conductivity and permittivity in the frequency range up to 1 MHz by standard LCR bridges. The conductivity signal responds to acids and salts, in particular to volcanic and Ammonium events. Permittivity is controlled by
the porosity and the density can be derived from permittivity measurements in the shallow part of an ice core. Especially for cores which are also analysed for impurities by other methods, DEP (and ECM) data are mainly used for synchronization purposes in order to provide a first timescale for an ice core, and to provide dielectric data for modelling synthetic radar profiles (Mojtabavi et al., 2022).

The DEP stratigraphy of the NGRIP, NEEM, and EGRIP cores was determined directly in the field. The DEP device used for
all three cores is described by Wilhelms (1996) and Wilhelms et al. (1998), the measuring procedure in detail in Mojtabavi et al. (2020). The spatial resolution is given by the 1-cm width of the moving capacitor plate, but measurements were made overlapping in 5 mm steps. The NGRIP cores were measured in a wide and varying range of frequencies between 500 Hz and up to 1 MHz but processed and further used are only the 250 kHz frequency data, which also is the case for the NEEM and EGRIP data. The DEP instrument was the first in the processing line, and due to lack of space and/or time in the trenches to
let the core temperatures equilibrate, the cores' temperatures varied during the day (as well as during the season). The DEP data are not corrected for this temperature variation. The data sets have been documented and made available as listed in the table in section 4.

For the GICC05 and GICC21 work, ECM was the main data set used for synchronization, but DEP conductivity data were used for synchronization of cores where there are gaps in the ECM data, and occasionally for supporting the annual-layer
identification across data gaps in the impurity records (Rasmussen et al., 2006; Sinnl et al., 2022).

### 3.3 Stable water isotopes, δ¹⁸O and δD

Water stable isotopes from DYE-3 and GRIP were used for annual-layer identification, while the accumulation rates at NGRIP and especially EGRIP are too low for the annual signal to survive diffusion in the top meters of the firn. An extensive amount



of stable oxygen isotope measurements $\delta^{18}O$ were carried out on the DYE-3 ice core just after drilling (1979–1981) (Dansgaard
et al., 1982): 63,000 $\delta^{18}O$ samples at a resolution of 8 samples per year or higher cover the period back to the year 5815 b2k
and the time interval from 6906 to 7898 b2k. Because the annual layer thicknesses thin with depth, the sample size had to be
decreased accordingly. The measurement plan aimed at collecting eight samples per year in order to resolve the annual cycle,
and used an ice-flow model to provide an a-priori estimate of annual layer thicknesses as a function of depth. For each drilled
ice core, the modelled annual layer was determined for the relevant depth, and the sample size was adjusted accordingly. In
practice, the samples sizes were cut after marks placed using an elastic band from Sigfus Johnsen's trousers as measuring
device: The elastic band had equidistant marks and was stretched in order to produce eight samples of equal size from each
ice section corresponding to the calculated layer thickness (Dill and Janke, 2013). Measurements were performed at the
Geophysical Isotope Laboratory (now part of the Niels Bohr Institute) in Copenhagen using the $CO_2$ equilibration method.
Vinther et al. (2006) added another 12,000 samples of $\delta D$ analyses from the periods 5816–6905 b2k and 7899–8313 b2k at a
resolution of 8 samples per year in order to complete the DYE-3 stable isotope from surface and back to the 8.2 ka cold event,
around which diffusion gradually smooths the signal to a degree where safe annual layer identification is no longer possible.
The $\delta D$ measurements were performed at the AMS [14]C Dating Centre at the University of Aarhus on a GV Instruments CF-
IRMS (Morrison et al., 2001). With this paper, we make the entire record available at the full, measured resolution. The $\delta^{18}O$
measurement uncertainty, estimated from repeated measurements of the same samples and comparisons to standards, is 0.1
per mil, while the $\delta D$ data uncertainty is 0.5 per mil.

Measurements on the DYE-3 4B and 18C shallow cores were performed using the same setup as described above for DYE-3
in 1983 and 1984, also aiming for an average resolution of 8 samples per year. The data were analysed by Clausen and Hammer
(1988) and Vinther et al. (2010), but have not been made publicly available before now.

For the GRIP core, $\delta18O$ measurements with a resolution of 2.5 cm are available back to 3845 b2k (Johnsen et al., 1997). This
resolution corresponds to 7–10 samples per year (with fewest samples per year in the earliest part of the record due to flow-
related thinning of the annual layers). The $\delta^{18}O$ measurements were made in 1989-90 in Copenhagen immediately after the
drilling. The uncertainty on the $\delta^{18}O$ values is 0.1 per mil. As diffusion in the firn and snow degrade the annual signal, the data
needs to be corrected for the effect of diffusion before being used for annual-layer identification (Vinther et al., 2006; Vinther
et al., 2010). Following the approach of Johnsen and Andersen (1997) and Johnsen et al. (2000), the deconvolution uses an
adaptive cut-off frequency permitting a fixed maximum amplification factor of 50 of the spectral components in the original
data. While the deconvolution generally restores or strengthens the annual signal, it may also create spurious peaks that can
complicate the correct identification of annual layers, in particular near melt layers.



### 3.4 Impurities measured on discrete samples by Ion Chromatography (IC)

For the NGRIP1 core, measurements of selected impurities ($Li^+$, $Na^+$, $NH_4^+$, $K^+$, $Mg^{2+}$, $Ca^{2+}$, $F^-$, methane sulfonate (MSA),

$Cl^-$, $NO_3^-$, and $SO_4^{2-}$) were made by Ion Chromatography (IC) for the past ~1800 years at a resolution of 5 cm, corresponding to about 4 samples per year on average. If only one parameter had been measured, this resolution would be marginal, but because different parameters have been measured and these peak at different times of the year, it is possible to identify annual layers with reasonably good accuracy using the records in combination. All samples were cut and decontaminated using a microtome knife in a laminar-flow bench at the field site. In 1996, the 9.850 – 349.250 meter depth interval was sampled

continuously in 5 cm resolution. In 1997, selected sections from the Holocene, mainly containing volcanic signals, were sampled in 5 cm or 2.5 cm depth resolution. Samples covering the 8.2 ka BP cold event were cut from the depth interval 1221-1237.5 m. The frozen samples were thawed in the laboratory and immediately after poured into precleaned sample vials for ion chromatography analyses. Samples from bags 212-403 (depth interval 116.05–221.65 m) were measured at the Department of Physical Geography, Stockholm University, Sweden. The rest of the samples were measured at the Geophysics Department,

Niels Bohr Institute, University of Copenhagen, Denmark. At both laboratories, the measurements where performed on a Dionex 500 IC. Data measured in Stockholm were calibrated to a calibration curve established using eight standards, while data measured in Copenhagen were calibrated to a linear calibration curve established using only a single standard. Ammonium concentrations measured in Copenhagen may be biased due to possible sample uptake of Ammonium from the air while in the liquid phase. In Stockholm, $Li^+$ was not measured and $F^-$-concentrations may be biased due to methods that were not optimal

for quantification of fast eluted ions. The uncertainty of the concentrations measured in Copenhagen is estimate to 10–15 % and come from a combination of bias from the solutions used for calibration, non-linearity in the measurements, and measurement drift during an analysis series. For extreme samples values, e.g. for large volcanic eruptions, the uncertainty can be larger than 15%. The concentrations measured in Stockholm are more accurate mainly due the use of more than one standard for calibration. Further information about analysis setup can be found in Littot et al. (2002), Siggaard-Andersen et al. (2002),

and Jonsell et al. (2007).

The sulphate data were released in connection with the study of (Plummer et al., 2012), where the volcanic and background sulphate levels were separated, but the remaining parameters have not yet been made publicly available.

### 3.5 Impurities measured by Continuous Flow Analysis (CFA)

Impurity records obtained by Continuous Flow Analysis (CFA) were pioneered by the University of Bern and are generally

very strong tools for annual-layer detection due to the often high resolution and because the annual layers are often detectable in several parallel records, sometimes with different seasonality. The first measurements performed by CFA on long ice-core intervals were those of the GRIP core (Fuhrer et al., 1993), comprising concentration profiles of $NH_4^+$, $Ca^{2+}$, and $H_2O_2$. The data were registered every 2 mm, but due to internal dispersion of the signal in the analytical setup, the shortest cycles that can be identified in the data are 2-3 cm. This limit is estimated from the response time of the system going from 90% to 10% of



of the amplitude when responding to a step concentration change (Röthlisberger et al., 2000). The achievable resolution depends on the selected melt speed (in the data sets provided here, it is typically 3.5-4 cm/min). Note that in some publications the resolution is defined by the respective e-folding time of the step signal, which is accordingly shorter (Erhardt et al., 2022). The data were used for annual-layer identification in the 7.9-10.3 ka b2k interval when creating GICC05 (Rasmussen et al., 2006), but data are also provided over GS-1 (roughly equivalent to the Younger Dryas) and GI-1 (roughly equivalent to the

Bølling-Allerød period). Formaldehyde was also measured but does not exhibit a usable annual signal and is not included in the data file. The measurement were performed as described in Fuhrer et al. (1993) and Sigg et al. (1994). Fuhrer et al. (1996) analysed the ammonium data, but the data set has remained unpublished in its full resolution until now.

The CFA measurement setup at the University of Bern was further developed, extended and refined, and used for obtaining records for several long ice cores. The NGRIP version of the Bern CFA system was operated in the field in 2000 and is

described in detail in Röthlisberger et al. (2000). In addition to the $NH_4^+$ and $Ca^{2+}$ measurements, which were already part of the GRIP setup, measurements of $Na^+$, $NO_3^-$, and $SO_4^{2-}$ were made, and a conductivity cell as well as an insoluble particle counter were also included. This NGRIP2 data set is only available from around 10.3 ka b2k and further back in time, and forms the backbone of GICC05 in large parts of the glacial, and is thoroughly described in Erhardt et al. (2022). The data file was released together with the Erhardt et al. (2022) paper, but due to larger analytical uncertainties and data quality issues, the

records of sulfate, dust particles, formaldehyde and hydrogen peroxide are not included. The $SO_4^{2-}$ data were nevertheless used by Lin et al. (2022), and the full-resolution sulfate data file back to 60 ka b2k is available as supplement to their paper.

Insoluble microparticles (mineral dust) was measured as part of the CFA setup, using a laser particle detector contributed by the University of Heidelberg and the Alfred Wegener Institute. The data were published separately by Ruth et al. (2003) and Ruth et al. (2007) as coarse-resolution data sets. A 5-cm resampled version of the data was also made available in connection

with the work of Gkinis et al. (2014), and annual values were released over short sections by Steffensen et al. (2008), but for annual-layer identification, the full resolution was used. The full-resolution data file is released with this work. The relative uncertainty of the laser particle measurements has been estimated to 5-10% for the number concentration and 15-20% for the mass concentration (Ruth et al., 2003), the latter including uncertainty in the size calibration. For the younger, more shallow parts of the NGRIP2 core, an impurity data set was measured at the Desert Research Institute in Reno covering 159.6 – 582.4

m depth (approximately from 730 to 3200 years b2k) with a resolution of 1 cm (Mcconnell et al., 2018), which has been used for GICC21.

For NEEM, the main data set used for the GICC21 work also comes from the Bernese CFA setup: the main NEEM data set is described and published together with the NGRIP2 data by Erhardt et al. (2022) and covers the entire core but with a relatively large rate of data loss in the brittle section. An additional CFA dataset measured by the Desert Research Institute related to
Sigl et al. (2013) is the record from the 400 m long shallow core NEEM-2011-S1, drilled about 100 m away from the NEEM main core, which was extended using NEEM main core ice in the 399−500 m interval by Sigl et al. (2015).

The limit of detection (LOD) of the CFA data (defined as 3 times the standard deviation of the signal baseline (derived from MillQ water) is typically 0.1 ppb for all CFA components (Erhardt et al., 2022; Röthlisberger et al., 2000). However, the reproducibility of calibration standards (1 sigma of their values) is much higher than the LOD and also affected by the
procedural blank to mix the standards. Thus, while the ice-core melt water stream is per se not affected by these effects, the translation of signal amplitudes to concentration using the calibration standards introduces this uncertainty. As a conservative estimate of this uncertainty in our concentration records, we use the uncertainty in the calibration standards (derived from a mean calibration curve on individual standards). This reproducibility (which is hence our estimated uncertainty of the concentration values) is 1 ppb for $Ca^{2+}$ and $Na^+$, 0.5 ppb for $NH_4^+$, 3 ppb for $NO_3^-$, and 1-4 ppb for $H_2O_2$ (Gfeller et al., 2014).
The relative uncertainties are dependent on the mean concentration of the respective species in the ice core, which are also time dependent. For example, for low $Ca^{2+}$ concentration values during the Holocene, the relative uncertainty can be 10-20 %, while for high glacial concentrations they are less than 2 %. For $Na^+$ a similar picture emerges with Holocene relative uncertainties of typically <5 % and glacial values < 1%. For $NH_4^+$ and $NO_3^-$, where the glacial/interglacial changes are small, the relative uncertainties are less than 10% and 2%, respectively (Gfeller et al., 2014). For $H_2O_2$, the relative uncertainty is
typically less than 2%.

Great care has been given to remove spurious values (for example from potential contamination at core breaks) in the records, but especially outlier values near data gaps need to be interpreted with extra care.

EGRIP CFA data are being prepared for publication by Erhardt et al. (in prep). The latter combines the CFA system already successfully deployed for the NEEM ice core but extended by an inductively-coupled plasma Time-Of-Flight Mass
Spectrometer (icpTOF, TOFWERK, Thun, Switzerland). The system allows the quantification of elemental concentrations over the full mass range 20-220 amu in millisecond resolution and, thus, allows us to detect individual dust particles. It also includes a Single Particle Extinction and Scattering instrument (SPES, EOS, Milano, Italy) which allows to quantify the diameter and refractive index of the dust particles in the ice. The data of these new instruments are still in the process of evaluation and are not provided here.

**3.6 Visual Stratigraphy (line scan data)**

A continuous high-resolution record of digital images was obtained from the NGRIP ice core in the depth interval 1330–3085 m during the 2000 and 2001 field seasons. The images are obtained as dark-field images from an indirect light source and provide detailed visual documentation of the ice core at high depth resolution. The visual stratigraphy grey-scale intensity profile (the line-scan profile) is obtained as an averaged intensity profile from the centre part of the stratigraphy images along
the direction of the core. The dataset covers the depth interval 1371.15-2425.00 m in 1 mm depth resolution. The dataset was



applied to construct the glacial part of the GICC05 ice-core chronology. The analytical techniques and the intensity profile are described in Svensson et al. (2005), to which we refer for a fuller description of uncertainties etc..

**3.7 GICC05 annual layer positions**

Based on the data described above, annual layers were identified for the construction of the Greenland Ice-Core Chronology
2005 (GICC05). The high-resolution version data file is released for the first time with this paper now that all the underlying data sets are also available. Previously, 10- and 20-year resolution data files containing the time scale and resampled $\delta^{18}O$ data have been released for different time intervals together with relevant dating papers. The data set consists of the location of the annual markers in the GICC05 time scale for each core's depth sections where data were available and sufficiently resolved to allow annual dating. The markers are placed in the winter or spring depending on the availability of data (e.g. using the winter
$\delta^{18}O$ minimum, winter Sodium concentration maximum, spring dust/Calcium concentration maximum, or line-scan grey-scale peaks in the deepest parts). Across data gaps, markers are placed by interpolation assuming a constant layer thickness or using other impurity species with different seasonality (e.g. using summer Ammonium or Nitrate peaks). Therefore, the criteria for where the annual markers are placed vary between sections, and care should be taken when interpreting data on annual scale. The dating of the 0–7.9 ka b2k part is based mainly on isotope data and is described by Vinther et al. (2006). Impurity records
from GRIP and NGRIP2 form the basis for the multi-parameter annual layer identification across the 7.9–14.7 ka b2k interval (Rasmussen et al., 2006). In the deeper parts, the line scan profile plays a larger role together with the best-resolved of the CFA parameters. The dating of the 14.7–41.8 ka b2k part is described in Andersen et al. (2006), while the details of the 41.8–60.0 ka b2k part can be found in Svensson et al. (2008). When counting layers, uncertainty is introduced when an annual layer is backed up by evidence only in some of the data series, or when a certain well-resolved feature is suspected to contain more
than one annual layer. The cases of ambiguity in the annual layer identification process have been marked using so-called uncertain layer markings. These uncertain layer markings were included in the time scale as ½ ± ½ years, with the ± ½ years forming the basis for quantifying the so-called maximum counting error. The concept of maximum counting error is further discussed in Rasmussen et al. (2006). If needed, the maximum counting error can in a standard deviation context be regarded as an approximation of the 2σ uncertainty (Andersen et al., 2006).
In the Holocene, GS-1, and GI-2, the published time scale was derived from annual layer markings by manually determining which half of the uncertain layer markings to count as years (denoted "type 1"), and which to skip (denoted "type 2"). With the data file, we provide a separate table detailing which uncertain layers were assigned to each of these two cateories. The maximum counting error was estimated from the number of uncertain layer markings as a constant relative uncertainty for each period with similar data availability and characteristics: 21–3,845 a b2k (0.25%), 3,846–6,905 a b2k (0.5%), 6,906–
10,276 a b2k (2%), 10,277–11,703 a b2k (0.67%), 11,703–12,896 a b2k (3,3%), 12,896–14,075 a b2k (2.6%), 14,075–14,692 a b2k (2.7%) (see table 2 in Vinther et al. (2006), and table 3 in Rasmussen et al. (2006)). From GS-2 and below every 2nd uncertain layer was counted as a year and the maximum counting uncertainty increased by one year, giving rise to a variable



relative counting error ranging from 4% in the warm interstadial periods to 7% in the cold stadials, and averaging 5.3% (Andersen et al., 2006; Svensson et al., 2008).


**Data availability**

| Parameter | Site | References | Link/doi |
|---|---|---|---|
| ECM | DYE-3 main core | This work | https://doi.org/10.1594/PANGAEA.942849<br>Temporary link for review (common link for the three DYE-3 ECM data sets):<br>https://www.pangaea.de/tok/5e1cd34a4b5e89a1c44bacb72c13b4f37d401b47 |
| | DYE-3 4B | This work | https://doi.org/10.1594/PANGAEA.942843<br>Temporary link for review (common link for the three DYE-3 ECM data sets):<br>https://www.pangaea.de/tok/5e1cd34a4b5e89a1c44bacb72c13b4f37d401b47 |
| | DYE-3 18C | This work | https://doi.org/10.1594/PANGAEA.942847<br>Temporary link for review (common link for the three DYE-3 ECM data sets):<br>https://www.pangaea.de/tok/5e1cd34a4b5e89a1c44bacb72c13b4f37d401b47 |
| | GRIP | This work | https://doi.org/10.1594/PANGAEA.942944<br>Temporary link for review:<br>https://www.pangaea.de/tok/b0b8821b04a2edc52b1702bd67b00e2208ac025b |
| | NGRIP1 and NGRIP2 | Rasmussen et al. (2013) | https://www.iceandclimate.nbi.ku.dk/data/ngrip2013ecm.txt, mirrored at https://doi.org/10.1594/PANGAEA.831528 and at WDC Paleo |
| | NEEM | Rasmussen et al. (2013) | https://www.iceandclimate.nbi.ku.dk/data/neem2013ecm.txt, mirrored at https://doi.org/10.1594/PANGAEA.831528 and https://doi.org/10.25921/gab6-fa09 |
| | EGRIP, down to 1383.84 m depth | Mojtabavi et al. (2020) | https://doi.org/10.1594/PANGAEA.922199 |
| DEP | NGRIP1, down to 1372 m depth | Mojtabavi et al. (2020) | https://doi.org/10.1594/PANGAEA.922191 |



| | NGRIP2, down to 1298.55 m depth | Mojtabavi et al. (2022) | https://doi.org/10.1594/PANGAEA.922306 |
|---|---|---|---|
| | NGRIP2, from 1298.7 m depth | Rasmussen et al. (2013) | https://www.iceandclimate.nbi.ku.dk/data/ngrip2013dep.txt and at WDC Paleo |
| | NEEM, down to 1493.297 m | Mojtabavi et al. (2020) | https://doi.org/10.1594/PANGAEA.922193 |
| | EGRIP, down to 1383.84 m depth | Mojtabavi et al. (2020) | https://doi.org/10.1594/PANGAEA.919313 |
| Water stable isotopes | DYE-3 main core | This work | https://doi.org/10.1594/PANGAEA.942945<br>Temporary link for review:<br>https://www.pangaea.de/tok/8c79195d8f4a9eae2d6a4ab51d791054510adee9 |
| | DYE-3 4B | This work | https://doi.org/10.1594/PANGAEA.942751<br>Temporary link for review:<br>https://www.pangaea.de/tok/5944c0168ecaeb78eb676c5aa3c340235a240136 |
| | DYE-3 18C | This work | https://doi.org/10.1594/PANGAEA.942937<br>Temporary link for review:<br>https://www.pangaea.de/tok/e6c800c6349d47c296a3dd65b19c13c9844af500 |
| | GRIP | This work | https://doi.org/10.1594/PANGAEA.942851<br>Temporary link for review:<br>https://www.pangaea.de/tok/a3e8f9e8be16d3daceff023cfcee9c792650130c |
| Soluble Impurities | NGRIP1, IC | This work (all parameters). Sulphate made available with Plummer et al. (2012) | https://doi.org/10.1594/PANGAEA.944172<br>Temporary link for review:<br>https://www.pangaea.de/tok/8f591eb20fb0c9f7b0208ef3e84c86f98cfcd748<br><br>https://www.iceandclimate.nbi.ku.dk/data/2012-12-03_NGRIP_SO4_5cm_Plummet_et_al_CP_2012.txt |

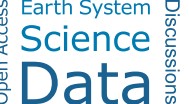



| | GRIP, CFA | Fuhrer et al. (1993), Sigg et al. (1994), Fuhrer et al. (1996), and this work | https://doi.org/10.1594/PANGAEA.942777 <br> Temporary link for review: <br> https://www.pangaea.de/tok/415ba47735cfe879651c292b2268a873c33549b1 |
|---|---|---|---|
| | NGRIP2, CFA, 1404 m downwards | Erhardt et al. (2022), Sulfate: Lin et al. (2022) | https://doi.org/10.1594/PANGAEA.935818. <br> Sulfate data as supplement to the paper at <br> https://cp.copernicus.org/articles/18/485/2022/ |
| | NGRIP2, CFA, 159.6–582.4 m | Mcconnell et al. (2018) | *[Joe McConnell has agree to make the data publicly available and we are working on getting a link to the data. The data are not ours, and are mentioned here for completeness only]* |
| | NEEM, CFA, all core | Erhardt et al. (2022) | https://doi.org/10.1594/PANGAEA.935837 |
| | NEEM, CFA, 399-500 m | Sigl et al. (2015) | Supplement to the paper at https://doi.org/10.1038/nature14565 |
| | NEEM-2011-S1 CFA | Sigl et al. (2013) | https://doi.org/10.18739/A2TH15 |
| | EGRIP, CFA | Erhardt et al. (in prep) | https://doi.org/10.1594/PANGAEA.945293 <br> *[Moratorium until 2023-03-15. The data file is related to the manuscript by Erhardt et al., and is not presented in this manuscript. It is mentioned here for completeness only]* |
| Dust | NGRIP2 | Ruth et al. (2003), Ruth et al. (2007), and this work | https://doi.org/10.1594/PANGAEA.945447 <br> Temporary link for review: <br> https://www.pangaea.de/tok/a0c2f9b2da6ea3231bc6bc585ba61001a81ca2ad |
| Grey-scale profile from the visual | NGRIP | Svensson et al. (2005) | https://doi.pangaea.de/10.1594/PANGAEA.941174 |




| stratigraphy scans | | | |
|---|---|---|---|
| GICC05 annuals | DYE-3, GRIP, NGRIP1, NGRIP2 | This work and, depending on age range used, Vinther et al. (2006), Rasmussen et al. (2006), Andersen et al. (2006), and/or Svensson et al. (2008). | https://doi.org/10.1594/PANGAEA.943195<br>Temporary link for review:<br>https://www.pangaea.de/tok/023c71d6caa354bd56a7e51c48aca102f823d7c1 |
| GICC21 annuals | EGRIP, NEEM, NGRIP1, NGRIP2, NEEM-2011-S1, GRIP, DYE-3 (main, 4B, and 18C) | Sinnl et al. (2022) | Supplement to the paper at https://cp.copernicus.org/articles/18/1125/2022/ |

**Conclusions**

With this paper and the associated data sets, the data underpinning the GICC05 and GICC21 time scales have been made available. We hope that this will stimulate further work on high-precision ice-core chronologies, both in relation to the
revision of GICC05, which has only just started with the publication of the GICC21 for the first 3.8 ka b2k (Sinnl et al., 2022), and in other contexts. In view of the complicated data matrix spanning a wide range of parameters, depth intervals, ice cores and decades of method development, we encourage future users of the data to get in contact with the respective research groups that measured the data to obtain expert advice on the data quality and its limitation for specific applications.



**Author contributions**

SOR compiled the metadata for the non-published data sets and carried out the Pangaea data review process in collaboration with co-authors (see each data file for names), except for the line-scan data file, which was publicised by AS. SOR wrote the manuscript with contributions and feedback from all co-authors.

**Competing interests**

The authors declare that they have no conflict of interest.

**Acknowledgements**

We thank all people involved in decades of drilling, ice-core processing and analysis, logistics, and keeping the field camps running, and all the people who have worked in the laboratories and with data processing.

DYE-3 was supported financially and logistically by the U.S. National Science Foundation (DPP), the Swiss National Science Foundation, the Commission for Scientific Research in Greenland, and the Danish Natural Science Research Council.

GRIP was organized by the European Science Foundation with contributions from the national science foundations in Belgium, Denmark, France, Germany, Iceland, Italy, Switzerland, and the United Kingdom. We thank the XII Directorate of CEC, the Carlsberg Foundation, the Commission for Scientific Research in Greenland, and the University of Iceland Research Foundation for financial support.

NGRIP was directed and organized by the Department of Geophysics at the Niels Bohr Institute for Astronomy, Physics and

Geophysics, University of Copenhagen. It was supported by funding agencies in Denmark (SNF), Belgium (FNRS-CFB), France (IPEV and INSU/CNRS), Germany (AWI), Iceland (RannIs), Japan (MEXT), Sweden (SPRS), Switzerland (SNF) and the USA (NSF, Office of Polar Programs).

NEEM was directed and organized by the Centre of Ice and Climate at the Niels Bohr Institute and US NSF, Office of Polar Programs. It was supported by funding agencies and institutions in Belgium (FNRS-CFB and FWO), Canada (NRCan/GSC),

China (CAS), Denmark (FIST), France (IPEV, CNRS/INSU, CEA and ANR), Germany (AWI), Iceland (RannIs), Japan (NIPR), South Korea (KOPRI), The Netherlands (NWO/ALW), Sweden (VR), Switzerland (SNF), the United Kingdom (NERC) and the USA (US NSF, Office of Polar Programs) and the EU Seventh Framework programmes Past4Future and WaterundertheIce.

EGRIP is directed and organized by the Centre for Ice and Climate at the Niels Bohr Institute, University of Copenhagen. It is

supported by funding agencies and institutions in Denmark (A. P. Møller Foundation, University of Copenhagen), USA (US National Science Foundation, Office of Polar Programs), Germany (Alfred Wegener Institute, Helmholtz Centre for Polar and Marine Research), Japan (National Institute of Polar Research and Arctic Challenge for Sustainability), Norway (University of Bergen and Bergen Research Foundation), Switzerland (Swiss National Science Foundation), France (French Polar Institute



Paul-Emile Victor, Institute for Geosciences and Environmental research) and China (Chinese Academy of Sciences and
Beijing Normal University).

A lot of the work generating the data presented here was supported by research grants from private and public foundations. We wish in particular to acknowledge support from the Carlsberg Foundation for supporting the work on GICC05 and GICC21, the latter as part of the project ChronoClimate.




## Figures

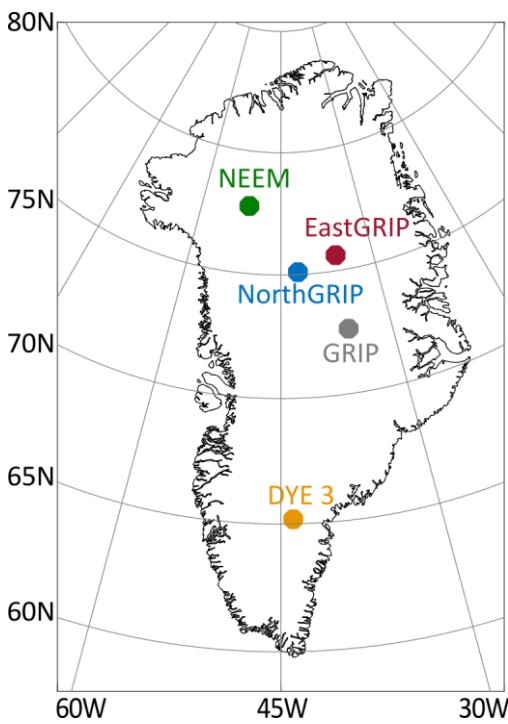

Figure 1: Map of Greenland with core locations.

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
