# Peer review of "Ice-core data used for the construction of the Greenland Ice-Core Chronology 2005 and 2021 (GICC05 and GICC21)"

_Earth System Science Data, 2022_

## Author Response (AR1)

We thank Eric Wolff, an anonymous reviewer, and the editor for their comments and efforts.

Reviews below are shown in italics with comments in normal typeface. We have made all changes according to the online discussion.

**Review by RC1 Eric Wolff**

*Major comment: Dye 3 ECM. Somehow the number of significant figures in both depth and acidity have been reduced to the point where the data are unuseable (depths given only as metres so that many data have the same depth). Please revise this dataset.*

> Fixed by PANGAEA

*Additional comment on data: The file providing the annual layer depths is done for GICC05, but for GICC21 we are simply referred to the supplement to Sinnl et al (2022). This is OK but it leaves the reader who has found the GICC05 layers unaware that for the last 3.8 kyr they have been superceded. I realise this may be hard to do but wouldn't it be valuable to add a column giving the corresponding depths (even if only for 1 site) for GICC21 until 3.8 ka b2k, so that the reader knows they are better to use? This is just a question of copying some columns from that supplement and would have the added value of making the GICC21 data available in a more regular database.*

> Citing the previous online reply: GICC21 applies to more cores than GICC05, and I fear that the data file will be very unhandy to use if we merge GICC05 with GICC21 layer positions plus the other supporting material that comes with GICC21. I suggest that we make a clear reference in the metadata of the GICC05 annual layer to the location of the corresponding GICC21 spreadsheet which has both the annual layer positions and a GICC05-GICC21 transfer table. Then all users will know that a newer time scale is available.

> We have requested to PANGAEA that the relevant comment is added to the metadata.

*Line 25,26: it would be useful to include in the abstract that GICC05 is to 60 ka and GICC21 so far to 3.8 ka.*

> Added as suggested. The sentence now reads: "The data series were used for counting annual layers 60,000 years back in time during the construction of the Greenland Ice-Core Chronology 2005 (GICC05) and/or the revised GICC21, which currently only reaches 3,800 years back."

*Line 166 "discretely" should be "discrete".*

> Fixed, thank you.

*Line 195. I think ueq or uequiv but no "." would be normal. I do wonder why ECM data are supplied as H+ (which the paper admits is uncalibrated) rather than as what was measured, ie current. Possibly this is simply a question of what level of product has been stored but maybe it is worth a comment?*

Unit fixed. Sentence added: "Despite the tentative calibration, we provide the data as [H⁺] values as these were the ones stored when the data sets were originally processed."

*Line 204, 268, 355/7 etc "ammonium" not "Ammonium" unless it starts a sentence. Same for other ions, please check this or the copy editors might look out for it.*

Changed.

*Section 3.2. I guess they were not used in the layer counting but it might be worth mentioning for context that GRIP DEP data are available at 2 cm resolution at https://www.ncei.noaa.gov/access/paleo-search/study/17845*

Comment and reference added (line 218-219 in revised manuscript).

*Line 230. Come on, we need to see the elastic band! (no need to respond to this).*

*Line 268. I appreciate that the abstract to the dataset warns that CPH ammonium and STO lithium may be unreliable, but by the time one has downloaded the tab file as an Excel, that warning is lost. I feel it should be clearly mentioned in the metadata within the file; I also question the value of publishing data you believe to be wrong.*

Citing the online discussion: We do not think that the data are wrong, but as described in section 3.4, they may be subject to larger bias than for other species due to the analytical setup. The warning also appears on the PANGAEA data page and will this accompany the data in the downloaded Excel file.

*Line 292. You mention that Fuhrer et al analysed the ammonium data. Again for context would it be worth mentioning that Fuhrer et al (1999) analysed the Ca data? (Fuhrer, K., Wolff, E. W., and Johnsen, S. J.: Timescales for dust variability in the Greenland Ice Core Project (GRIP) ice core in the last 100,000 years, J. Geophys. Res., 104, 31043-31052, 1999.). Maybe you feel that because they didn't use the high resolution there is no relevance, and I would understand that.*

Added. Sentence now reads "Fuhrer et al. (1996) analysed the ammonium data and Fuhrer et al. (1999) analysed the calcium record, but the data set has remained unpublished in its full resolution until now."

*Line 310. McConnell spelling.*

Fixed

**Review RC3, Anonymous Referee #2**

Review in italics. We have made all changes according to the online discussion.

*An important output derived from the timescale are snow accumulation rates. For NEEM, these have been published in Rasmussen et al. 2013, but for GRIP and NGRIP Seierstadt et al. 2014 presented them in their paper, but did not provide them in their supplementary. When releasing such a large dataset of impurities measured in ice, many people may want to calculate fluxes etc. Hence, I would ask the authors to also provide the accumulation data from Seierstadt et al. This could also be done by just providing the thinning function for each core.*

> See online discussion. As a result, we have added section 3.8 which as a service provides some information on accumulation reconstructions

*L187: I am not sure the name of the laboratory technician is of relevance here? If this is an acknowledgement of her work, and she is fine with being named this is ok, but I suppose many other people could be named throughout the paper too.*

> Citing the online discussion: You are absolutely right that many people deserve acknowledgement for their work in the laboratories. The reason why we decided to explicitly mention Anita Boas is that we want to acknowledge that she played a particularly important role in the isotope measurements in Copenhagen, measuring several hundred thousand samples over decades with a never-failing eye for getting things right.

> We therefore keep the direct reference to Anita Boas.

*L230-232: Nice anecdote, but I guess "yeah, well, that's just, like, your opinion, man"? Please remove the reference to Dill & Janke.*

> I abide.

*L258: what do you consider to be "reasonably good"?*

> Citing the online discussion: The uncertainty of the counting is variable and published together with the GICC05 and GICC21 time scales. The point we are trying to make here is that the resolution of the IC data (3-4 samples per year, although a bit better in the top) would normally be at least marginal for annual layer detection, but that having several parallel series with different seasonality makes it possible. I have removed "with reasonably good accuracy" as this is a vague statement.

*Data-table: The linescan (greyscale) data is for NGRIP2, right? If so, please indicate this in the 2$^{nd}$ column of the table (just says "NGRIP").*

> Corrected. Thank you!

---

## Author Response (AR2)

[revised manuscript text omitted]
| | EGRIP, CFA | Erhardt et al. (2023) | https://doi.org/10.1594/PANGAEA.945293 |
| Dust | NGRIP2 | Ruth et al. (2003), Ruth et al. (2007), and this work | https://doi.org/10.1594/PANGAEA.945447 |
| Grey-scale profile from the visual stratigraphy scans | NGRIP2 | Svensson et al. (2005) | https://doi.pangaea.de/10.1594/PANGAEA.941174 |
| GICC05 annuals | DYE-3, GRIP, NGRIP1, NGRIP2 | This work and, depending on age range used, Vinther et al. (2006), Rasmussen et al. (2006), Andersen et al. (2006), and/or Svensson et al. (2008). | https://doi.org/10.1594/PANGAEA.943195 |
| GICC21 annuals | EGRIP, NEEM, NGRIP1, NGRIP2, NEEM-2011-S1, GRIP, DYE-3 (main, 4B, and 18C) | Sinnl et al. (2022) | Supplement to the paper at https://cp.copernicus.org/articles/18/1125/2022/ |

**Conclusions**

[revised manuscript text omitted]

Erhardt, T., many, other,Jensen, C. M., Borovinskaya, O., and Fischer, H.: Single Particle Characterization and authors:Total Elemental Concentration Measurements in Polar Ice Using Continuous Flow Analysis-Inductively Coupled Plasma Time-of-Flight Mass Spectrometry, Environmental Science & Technology, 53, 13275-13283, 10.1021/acs.est.9b03886, 2019.

Erhardt, T., Jensen, C. M., Adolphi, F., Kjær, H. A., Dallmayr, R., Twarloh, B., Behrens, M., Hirabayashi, M., Fukuda, K., Ogata, J., Burgay, F., Scoto, F., Crotti, I., Spagnesi, A., Maffezzoli, N., Segato, D., Paleari, C., Mekhaldi, F., Muscheler, R., Darfeuil, S., and Fischer, H.: 
[revised manuscript text omitted]